# Three-Dimensional Quantum Hall Effect in Topological Amorphous Metals

Jiong-Hao Wang[1] and Yong Xu[1,2]*

**1** Center for Quantum Information, IIIS, Tsinghua University, Beijing 100084, People's Republic of China
**2** Hefei National Laboratory, Hefei 230088, People's Republic of China
* yongxuphy@tsinghua.edu.cn

## Abstract

Weyl semimetals have been theoretically predicted to become topological metals with anomalous Hall conductivity in amorphous systems. However, measuring the anomalous Hall conductivity in realistic materials, particularly those with multiple pairs of Weyl points, is a significant challenge. If a system respects time-reversal symmetry, then the anomalous Hall conductivity even vanishes. As such, it remains an open question how to probe the Weyl band like topology in amorphous materials. Here, we theoretically demonstrate that, under magnetic fields, a topological metal slab in amorphous systems exhibits three-dimensional quantum Hall effect, even in time-reversal invariant systems, thereby providing a feasible approach to exploring Weyl band like topology in amorphous materials. We unveil the topological origin of the quantized Hall conductance by calculating the Bott index. The index is carried by broadened Landau levels with bulk states spatially localized except at critical transition energies. The topological property also results in edge states localized at distinct hinges on two opposite surfaces.

# 1 Introduction

Weyl semimetals [1–5], a celebrated example of gapless topological phases, have attracted considerable attention due to both fundamental interest in the emergent Weyl femions as well as novel topological properties and potential applications in future electronic devices [6–34]. Their energy bands touch linearly at a Weyl point protected by the Chern number over a two-dimensional surface enclosing the Weyl point in three-dimensional (3D) momentum space. By bulk-boundary correspondence, topological properties in Weyl semimetals manifest in the surface Fermi arcs connecting Weyl points with opposite chirality, which can give rise to *anomalous* Hall effect [7,8]. Remarkably, such Fermi arc states assisted by bulk chiral Landau levels in Weyl semimetals under magnetic fields can result in 3D quantum Hall effect [35–45].

Apart from crystalline materials with translational symmetry, nature also provides us with abundant amorphous solids without translational symmetry [46]. Although topological physics is established in crystalline materials, recent studies have suggested that amorphous systems can also display various topological phases [47–75]. In particular, it has been theoretically shown that Weyl semimetals without time-reversal symmetry (TRS) will become topological metals with nonzero anomalous Hall conductivity when lattice sites are positioned randomly [54]. However, it is significantly challenging to experimentally measure the anomalous Hall conductivity in realistic materials, especially for systems with multiple pairs of Weyl points. If a system respects TRS, then the anomalous Hall conductivity even vanishes in general. In addition, conventional methods to probe Weyl points and surface states using angle-resolved photoemission spectroscopy (ARPES) [15,17] may be impractical due to the absence of a well-defined momentum space. We therefore ask whether 3D quantum Hall effect can exist in amorphous Weyl materials under magnetic fields, which may provide a feasible approach to probing Weyl band like topology in amorphous materials.

In this work, we theoretically demonstrate that 3D quantum Hall effect can occur in amorphous systems by studying a tight-binding model on a random lattice under a magnetic field [see Fig. 1(a)]. We find the existence of quantized plateaus of the Hall conductance contributed by edge states [see Fig. 1(b)]. The edge states have a topological origin that can be characterized by the Bott index. Despite the presence of strong structural disorder, we show that the bulk states can still form Landau levels, albeit broadened. However, we show that these levels are localized, except at the critical transition point between two plateaus. As a consequence, only edge states contribute to the transport, ultimately resulting in the quantized Hall conductance. In a regular lattice, the 3D quantum Hall effect is explained through semiclassical analysis based on lattice momentum $k$ by showing that electrons cycle along the Fermi arcs on top and bottom surfaces aided by chiral Landau levels tunneling between the opposite surfaces [35,40]. In amorphous systems, while the translational symmetry is entirely lost so that the $k$ space description breaks down, we still observe the edge states localized at distinct hinges on opposite surfaces. Finally, we construct a time-reversal invariant model with double pairs of Weyl points when lattice sites are positioned regularly. In both regular and amorphous geometries, it has no *anomalous* Hall conductivity. However, when a magnetic field is applied, we see the appearance of nonzero quantized Hall conductance, demonstrating that 3D quantum Hall effect can be used as an indicator of topology for amorphous Weyl materials with vanishing *anomalous* Hall conductivity.

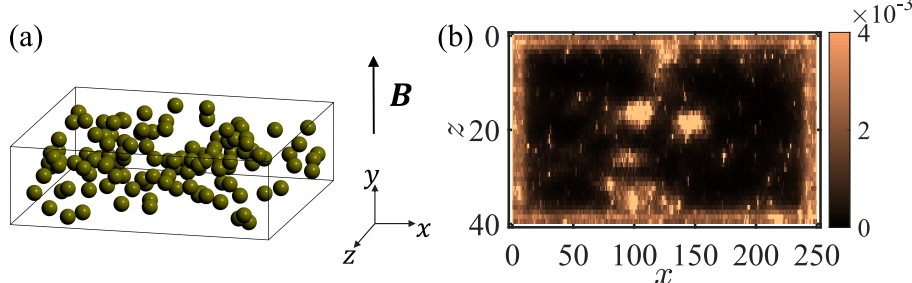

Figure 1: (a) Schematic of a random site configuration in a slab geometry with magnetic fields $\boldsymbol{B}$ along the $y$ direction. (b) Configuration averaged local density of states (DOS) summed over coordinates along $y$ for the Hamiltonian (1) on amorphous lattices with system size $L_x = 250$, $L_y = 20$, and $L_z = 40$ subjected to magnetic fields, illustrating the presence of edge states.

## 2   Model Hamiltonian

To demonstrate the existence of 3D quantum Hall effect in amorphous systems, we consider the following tight-binding Hamiltonian

$$\hat{H} = \sum_{\boldsymbol{r}} [\hat{c}_{\boldsymbol{r}}^{\dagger} T_0 \hat{c}_{\boldsymbol{r}} + \sum_{\boldsymbol{d}} t(|\boldsymbol{d}|) \hat{c}_{\boldsymbol{r}+\boldsymbol{d}}^{\dagger} T(\hat{\boldsymbol{d}}) \hat{c}_{\boldsymbol{r}}], \tag{1}$$

where $\hat{c}_{\boldsymbol{r}}^{\dagger} = (\hat{c}_{\boldsymbol{r},\uparrow}^{\dagger}, \hat{c}_{\boldsymbol{r},\downarrow}^{\dagger})$ with $\hat{c}_{\boldsymbol{r},\sigma}^{\dagger}$ being the fermionic creation operator of spin $\sigma$ at position $\boldsymbol{r}$, and $\hat{c}_{\boldsymbol{r}}$ being its Hermitian conjugate, the corresponding annihilation operator. $T_0 = M(k_w^2 - 6) \sigma_z + (4D_2 + 2D_1)\sigma_0$ is the onsite term, and $T(\hat{\boldsymbol{d}}) = M\sigma_z + i(\gamma/2)(\hat{d}_x \sigma_x + \hat{d}_y \sigma_y) - (D_2 \hat{d}_x^2 + D_1 \hat{d}_y^2 + D_2 \hat{d}_z^2)\sigma_0$ is the hopping matrix from the site at position $\boldsymbol{r}$ to a different site at $\boldsymbol{r} + \boldsymbol{d}$, where $\sigma_0$ is the identity matrix, $\sigma_{\nu}$ ($\nu = x, y, z$) are Pauli matrices, and $\hat{\boldsymbol{d}} = \boldsymbol{d}/|\boldsymbol{d}| = (\hat{d}_x, \hat{d}_y, \hat{d}_z)$ is the normalized separation vector. In the amorphous case, we set $t(|\boldsymbol{d}|) = \Theta(R_c - |\boldsymbol{d}|)e^{-\lambda(|\boldsymbol{d}|/a - 1)}$, decaying exponentially to simulate real materials. Here, $\Theta(R_c - |\boldsymbol{d}|)$ is the Heaviside step function such that hoppings for $|\boldsymbol{d}| > R_c$ are cut off and the lattice constant is set to $a = 1$. In a regular lattice with only nearest-neighbor hopping, i.e. the hopping strength $t(|\boldsymbol{d}|)$ set to be 1 for nearest neighbor and 0 otherwise, the Hamiltonian (1) reduces to a paradigmatic Weyl semimetal model corresponding to a momentum space Hamiltonian $H(\boldsymbol{k}) = M(k_w^2 - \boldsymbol{k}^2)\sigma_z + \gamma(k_x \sigma_x + k_y \sigma_y) + D_1 k_y^2 + D_2(k_x^2 + k_z^2)$ in the continuum limit, where $M$, $k_w$, $\gamma$, $D_1$ and $D_2$ are real system parameters. The Weyl points are located at $\boldsymbol{k} = (0, 0, \pm k_w)$ with energy $E_w = D_2 k_w^2$. When $D_1, D_2 \neq 0$, Fermi arcs are bent, which is essential to 3D quantum Hall effect when the Fermi energy lies at the energy of Weyl points [35, 40].

In the amorphous case, we consider sites that are positioned completely randomly in a slab, which is thin along $y$. To be explicit, the positions of lattice sites are sampled from uncorrelated uniform distributions within the interval $[0, L_{\nu})$, where $L_{\nu}$ is the length of the system along $\nu$ with $\nu = x, y, z$. We take the total number of sites as $N = \rho V$ with the volume of the system $V = L_x L_y L_z$ and the average density $\rho = 1$. An external magnetic field $\boldsymbol{B} = (0, B, 0)$ is applied in the $y$ direction [see Fig. 1(a)]. The magnetic field modifies the hopping matrix $T(\hat{\boldsymbol{d}})$ by a phase factor $e^{-i\frac{e}{\hbar}\int_{\boldsymbol{r}}^{\boldsymbol{r}+\boldsymbol{d}} \boldsymbol{A} \cdot d\boldsymbol{r}'}$ where the physical constants are set to be $e = \hbar = 1$ and we choose a gauge such that the vector potential is $\boldsymbol{A} = (Bz, 0, 0)$. Note that without changing the physics qualitatively, we follow previous works to ignore the contribution of the Zeeman effect [35, 40]. In numerical computations throughout the paper, we take $M = 8, A = 50, k_w = 1.5, D_1 = 1, D_2 = 4, R_c = 2.5, B = \pi/22$ and $\lambda = 3$ without

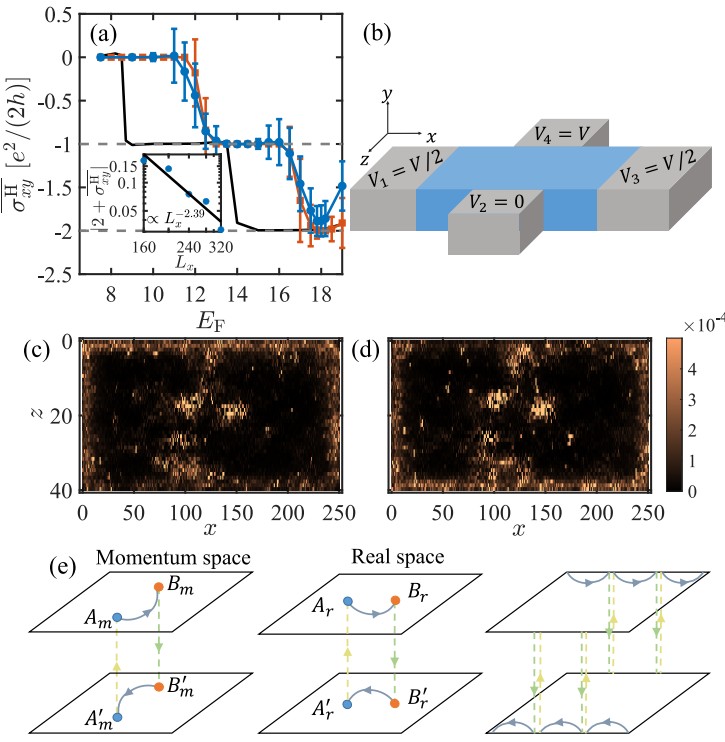

Figure 2: (a) Sample averaged Hall conductance $\overline{\sigma^{\mathrm{H}}_{xy}}$ (blue line) and Bott index (red line) versus the Fermi energy $E_{\mathrm{F}}$ of the Hamiltonian (1) under magnetic fields on amorphous lattices compared with the Hall conductance on regular lattices (black line). The system size for the Hall conductance is $L_x = 200, L_y = 20, L_z = 40$ and that for the Bott index is $L_x = 25, L_y = 20, L_z = 44$. Inset: the Hall conductance at $E_{\mathrm{F}} = 18.2$ for $L_x = 160, 200, 240, 280, 320$ (blue points) fitted by a black line in the logarithmic scale, exhibiting the power-law decay. We keep the ratio $L_x/L_z = 5$, and $L_y = 20$ is fixed. (b) Schematic illustration of a four-terminal setup to extract the Hall conductance. (c)-(d) Sample averaged local DOS on the top and bottom layer, respectively, calculated for $L_x = 250, L_y = 20, L_z = 40$ within $L_y - 1 \leq z \leq L_y$ and $0 \leq z \leq 1$ (e) Schematic of cyclic trajectories of an electron under magnetic fields [in momentum space (left panel) and real space (middle panel)] and the resulting skipping edge states (right panel).

loss of generality, and the results are averaged over 100 sample configurations unless stated otherwise.

In a regular lattice, the Hamiltonian (1) hosts Fermi arcs on top and bottom surfaces viewed in the $y$ direction. The Fermi arcs contribute to edge channels with the assistance of bulk chiral Landau levels under magnetic fields, giving rise to 3D quantum Hall effect [35, 40]. In the amorphous case, while previous study suggests the existence of surface states despite the absence of translational symmetry [54], it is unclear whether isolated Landau levels can survive so as to result in 3D quantum Hall effect. To investigate the possibility, we will calculate the Hall conductance, the Bott index, the DOS and the localization properties of the Hamiltonian (1) in random site configurations.

# 3  Hall conductance, Bott index and edge states

To show the presence of 3D quantum Hall effect in amorphous systems, we calculate the Hall conductance using Landauer-Büttiker formula under four-terminal measurement with voltage $V_1 = V_3 = V/2, V_4 = V$ and $V_2 = 0$ applied on ideal leads, as shown in Fig. 2(b). The Hall conductance $\sigma_{xy}^{\mathrm{H}}$ is given by [54]

$$\sigma_{xy}^{\mathrm{H}} = \frac{e^2}{2h}(T_{14} - T_{12}), \tag{2}$$

where $T_{ij}$ is the transmission probability from lead $j$ to $i$ computed numerically using non-equilibrium Green's function method [76–79]. In the phase with 3D quantum Hall effect, $\sigma_{xy}^{\mathrm{H}}$ is quantized to an integer multiple of $e^2/(2h)$. Note that in the quantized region, the Hall conductance determined by Eq. (2) is identical to that measured under constant longitudinal and no transverse current up to a multiplier $1/2$ [35, 40].

In Fig. 2(a), we plot the sample averaged Hall conductance $\overline{\sigma_{xy}^{\mathrm{H}}}$ with respect to the Fermi energy $E_F$ (blue line). Remarkably, we see a plateau quantized at $\overline{\sigma_{xy}^{\mathrm{H}}} = -e^2/(2h)$ in the region $13 \lesssim E_F \lesssim 16$, establishing the presence of amorphous 3D quantum Hall effect. Specifically, as $E_F$ is increased from $E_F = 7.5$, we see $\overline{\sigma_{xy}^{\mathrm{H}}} = 0$ for $E_F \lesssim 11$ and then $\overline{\sigma_{xy}^{\mathrm{H}}}$ undergoes a transition around $E_F \approx 12$ into the plateau at $\overline{\sigma_{xy}^{\mathrm{H}}} = -e^2/(2h)$. As we further raise $E_F$, $\overline{\sigma_{xy}^{\mathrm{H}}}$ declines and reaches a plateau near $\overline{\sigma_{xy}^{\mathrm{H}}} = -2e^2/(2h)$ for $17.8 \lesssim E_F \lesssim 18.2$. Our numerical calculations further show that $|2 + \overline{\sigma_{xy}^{\mathrm{H}}}| \propto L_x^{-2.39}$ with system size $L_x$ at $E_F = 18.2$ [see the inset of Fig. 2(a)], suggesting that within this region, $\overline{\sigma_{xy}^{\mathrm{H}}}$ can reach the quantized value of $-2e^2/(2h)$ in the thermodynamic limit. Compared with the Hall conductance $\overline{\sigma_{xy}^{\mathrm{H}}}$ of a system with regular lattice sites (black line), we see that the corresponding plateaus for the amorphous system move toward higher $E_F$, which is attributed to the shift of Landau levels in the amorphous case as will be discussed later.

To reveal the topological nature of 3D quantum Hall effect, we calculate the Bott index [80, 81] defined as

$$\mathrm{Bott} = \frac{1}{2\pi}\mathrm{Im}\mathrm{Tr}\log(U_z U_x U_z^\dagger U_x^\dagger), \tag{3}$$

where $[U_\nu]_{mn} = \langle\psi_m|e^{i2\pi\hat{\nu}/L_\nu}|\psi_n\rangle$ ($\nu = z, x$) with $\hat{\nu}$ being position operators along $\nu$ and $|\psi_n\rangle$ ($1, 2, ..., N_{occ}$) being occupied single-particle states under $E_F$. In our numerical computations, we impose periodic boundary conditions (PBCs) along $x$ and $z$ and open boundary conditions (OBCs) along $y$, given the fact that the surface states on top and bottom surfaces are crucial to the 3D quantum Hall effect.

Figure 2(a) shows the computed Bott index (red line), which agrees well with the Hall conductance (blue line). In fact, the Bott index is less affected by finite-size effects due to the imposed PBCs. To be specific, for $17.8 \lesssim E_F \lesssim 18.2$, the Bott index is well quantized at $\mathrm{Bott} = -2$, further verifying the existence of the plateau with $\overline{\sigma_{xy}^{\mathrm{H}}} = -2e^2/(2h)$ in thermo-dynamic limit. In addition, within the transition region between quantized plateaus, the Bott index exhibits a sharper decline than the Hall conductance. The consistence between the Bott index and the Hall conductance thus establishes the topological origin of the 3D quantum Hall effect.

Because of the topological properties, we expect the existence of hinge states on top and bottom surfaces, leading to the 3D quantum Hall effect. To identify the hinge states, we calculate the local DOS at energy $E$ and position $\boldsymbol{r}$ based on

$$\rho(\boldsymbol{r}, E) = -\frac{1}{\pi}\mathrm{Im}\mathrm{Tr}\, G_{\boldsymbol{rr}}(E) \tag{4}$$

using the recursive Green's function method [76, 77]. Here, $G_{\boldsymbol{rr}}(E) = \langle \boldsymbol{r}|(E-H+i0^+)^{-1}|\boldsymbol{r}\rangle$ where $H$ is the single-particle first quantization form of the Hamiltonian (1) under OBCs along all three directions, $0^+$ denotes a positive infinitesimal, and $|\boldsymbol{r}\rangle$ is the site state at position $\boldsymbol{r}$. Figure 1(b) shows the local DOS at $E = 14$ summed over the coordinates along $y$, remarkably exhibiting the existence of edge states despite the presence of strong structural disorder. Note that the small bright regions in the interior are contributed by localized bulk states. Ref. [54] shows that a Weyl semimetal develops into a topological metal in a random lattice with surface states. Such edge states on top and bottom surfaces thus arise from these surface states under magnetic fields.

Figure 2(c) and (d) further display the local DOS at the top and bottom layer along $y$, respectively. It surprisingly illustrates that the hinge states are primarily concentrated at the $z = 0$ edge on the top surface [Fig. 2(c)] while at the other edge on the bottom surface [Fig. 2(d)]. In a regular lattice, such a distribution of hinge states on opposite surfaces arises from Fermi arc states assisted by bulk chiral Landau levels [35, 40]. Specifically, according to the semiclassical equations of motion, $\dot{\boldsymbol{r}} = \partial \varepsilon_{\boldsymbol{k}}/(\hbar \partial \boldsymbol{k})$ and $\hbar \dot{\boldsymbol{k}} = -e\dot{\boldsymbol{r}} \times \boldsymbol{B}$ [82], if an electron starts at $A_m$ in momentum space and $A_r$ in real space, then under magnetic fields, it will move from $A_m$ to $B_m$ in momentum space following the Fermi arc. Meanwhile, the electron moves from $A_r$ to $B_r$ in real space [see Fig. 2(e)]. At $B_m$, the electron tunnels to $B'_m$ on the opposite surface through a bulk chiral Landau level. In real space, the electron tunnels to $B'_r$ on the opposite surface. On the bottom surface, the electron further moves from $B'_m$ ($B'_r$) to $A'_m$ ($A'_r$) in momentum (real) space, completing cyclic motions in both momentum and real spaces. Such cyclic motions form Landau levels. However, at the edges of the top and bottom surfaces, such cyclic motions are not allowed, leading to skipping edge states. While the semiclassical analysis is based on $\boldsymbol{k}$ space for translational-invariant systems, our results show the presence of the distinct profile of the local DOS for edge states on opposite surfaces in amorphous systems.

## 4 Landau levels and localization properties

To further elaborate on the mechanism of 3D quantum Hall effect, we compute the DOS defined as $\rho(E) = \sum_i \delta(E - E_i)/(2N)$, where $E_i$ is the $i$th eigenvalue of the Hamiltonian and $N = L_x L_y L_z$. We use the kernel polynomial method (KPM) to compute the DOS with Chebyshev expansion up to order $N_c$ [83] under the same boundary conditions as for the Bott index. To study the localization properties, we calculate the localization length of the system in a long bar geometry [84, 85] by

$$\lambda^{-1}(E) = -\lim_{L_z \to \infty} \frac{1}{2L_z} \log \text{Tr}\,[G_{1z}(E)G_{1z}^\dagger(E)], \tag{5}$$

where $G_{1z}(E) = \langle 1|(E-H+i0^+)^{-1}|z\rangle$ is the submatrix of the Green's function between the site states in the first and final layer along $z$, which can be obtained iteratively after partitioning the bar into layers each with thickness $R_c$ along $z$ [76, 77]. In the calculations, we apply PBCs along $x$ and OBCs along $y$ and $z$.

Figure 3(a) displays the sample averaged DOS $\overline{\rho}(E)$ as a function of energy $E$ (blue line), remarkably showing the existence of Landau levels despite the presence of strong structural disorder in amorphous lattices. In stark contrast to the regular case (black line), the Landau levels are broadened, resulting in an overlap between neighboring levels. The system thus cannot form band insulators for the bulk states as in the regular case. Fortunately, these states are spatially localized as reflected by the decrease of the averaged normalized localization length $\overline{\Lambda} = \overline{\lambda}/L_x$ with the increasing of system sizes [see Fig. 3(b)] [84, 85]. Since the localized states have no contribution to electrical transport, the quantized Hall conductance is attributed

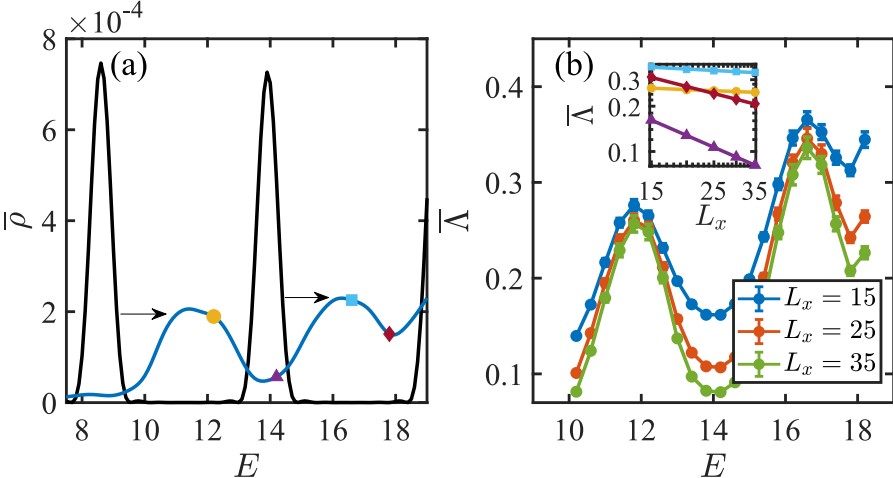

Figure 3: (a) Sample averaged DOS $\overline{\rho}(E)$ versus the energy $E$ for amorphous systems (blue line) and regular lattices (black line), computed by the KPM with the expansion order $N_c = 2^{13}$. Here, $L_x = 25$, $L_y = 20$, and $L_z = 44$. Arrows indicate the shifting direction of Landau levels induced by structural disorder. (b) Sample averaged normalized localization length $\bar{\Lambda}$ with respect to $E$ for $L_x = 15, 25$ and 35. Here $L_y = 20$ and $L_z = 10^4$. The inset shows the finite-size scaling of $\bar{\Lambda}$ versus $L_x$ at $E = 12.2$, 14.2, 16.6 and 17.8 [highlighted by yellow circle, purple triangle, blue square and red diamond in (a), respectively]. For $E = 14.2$ and 17.8, we take $L_z = 10^4$ and $L_y = 20$, and for $E = 12.2$ and 16.6, we take $L_z = 10^5$ and $L_y = 20$ (the results are averaged over 40 samples).

to the edge states (see the following discussion). Figure 3(b) also suggests that around the peaks of the two Landau levels, $\overline{\Lambda}$ converges to finite values as the system enlarges, signaling critical extended behaviors. Such critical points correspond to the transition points between neighboring plateaus of the Hall conductance. Specifically, we fit $\overline{\Lambda}$ computed for different system sizes as $\overline{\Lambda} \propto L_x^{-\alpha}$ at $E = 12.2$ (yellow circle), 14.2 (purple triangle), 16.6 (blue square) and 17.8 (red diamond), yielding $\alpha = 0.07, 0.82, 0.09$ and 0.48, respectively [see the inset of Fig. 3(b)]. The nonzero slope $\alpha$ suggests the localized character. For $E = 12.2$ and 16.6, the slope $\alpha$ is very small, indicating their closeness to critical energies. Note that the critical energies are difficult to determine accurately in our 3D case due to the large consumption of computational resources.

Figure 3(a) also illustrates that the Landau levels in our amorphous lattices move to higher energy compared with its regular counterpart (black line), explaining the up-moving behavior of the Hall conductance plateau mentioned above .

## 5   Model with TRS

To demonstrate that 3D quantum Hall effect can be utilized to detect the Weyl band like topological properties in amorphous systems with TRS, we construct the following Hamiltonian,

$$\hat{H}_T = \sum_{\boldsymbol{r}}[\hat{c}_{\boldsymbol{r}}^{\dagger}T_T^0\hat{c}_{\boldsymbol{r}} + \sum_{\boldsymbol{d}}t(|\boldsymbol{d}|)\hat{c}_{\boldsymbol{r}+\boldsymbol{d}}^{\dagger}T_T(\hat{\boldsymbol{d}})\hat{c}_{\boldsymbol{r}}] \tag{6}$$

with four internal degrees of freedom at each site, i.e. $\hat{c}_{\boldsymbol{r}}^{\dagger} = (\hat{c}_{\boldsymbol{r},1}^{\dagger}, \hat{c}_{\boldsymbol{r},2}^{\dagger}, \hat{c}_{\boldsymbol{r},3}^{\dagger}, \hat{c}_{\boldsymbol{r},4}^{\dagger})$. Here $T_T^0 = M(k_w^2 -6)\tau_z\sigma_0 + (4D_2 + 2D_1)\tau_0\sigma_0 + \beta\tau_y\sigma_y$ and $T_T(\hat{\boldsymbol{d}}) = M\tau_z\sigma_0 + (i\gamma/2)(\hat{d}_x\tau_x\sigma_z + \hat{d}_y\tau_y\sigma_0) - (D_2\hat{d}_x^2 +$

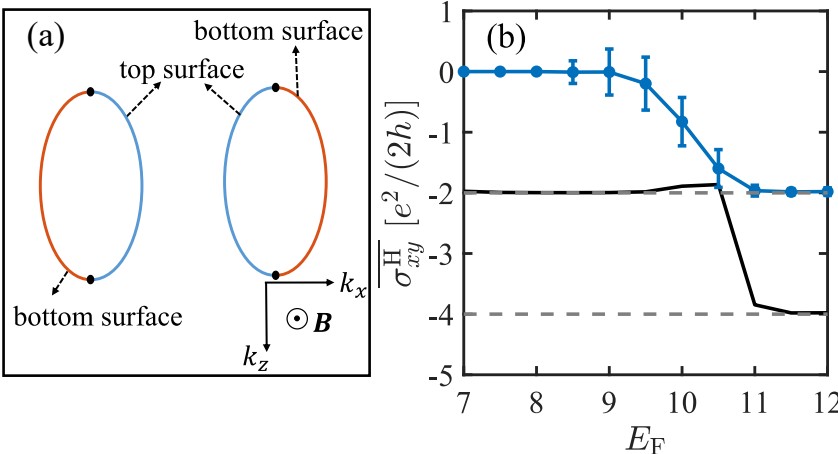

Figure 4: (a) Schematic of the Weyl points and Fermi arcs for the Hamiltonian (6) with TRS on regular lattices. (b) Configuration averaged Hall conductance $\overline{\sigma^{\mathrm{H}}_{xy}}$ versus the Fermi energy $E_{\mathrm{F}}$ for the Hamiltonian (6) on amorphous lattices with system size $L_x = 250, L_y = 20$ and $L_z = 40$ (blue line) compared with the Hall conductance for regular lattices with $L_x = 350, L_y = 20$ and $L_z = 40$ (black line). Here, we take $M = 13, \alpha = -80, \beta = 20, B = 1/12$ and other parameters are the same as before.

$D_1 \hat{d}_y^2 + D_2 \hat{d}_z^2)\tau_0\sigma_0 + i\alpha\hat{d}_y\tau_x\sigma_y/2$, with $\tau_x$ ($\nu = x, y, z$) being another set of Pauli matrices referring to orbital degrees of freedom. The Hamiltonian respects TRS, i.e. $\hat{T}\hat{H}_T\hat{T}^{-1} = \hat{H}_T$ with time-reversal operator $\hat{T}$ acting as $\hat{T}i\hat{T}^{-1} = -i$ and $\hat{T}\hat{c}_{\boldsymbol{r}}\hat{T}^{-1} = i\tau_0\sigma_y\hat{c}_{\boldsymbol{r}}$.

For a regular lattice, the Hamiltonian $\hat{H}_T$ describes a Weyl semimetal with double pairs of Weyl points and two pairs of Fermi arcs, as shown schematically in Fig. 4(a). Without magnetic fields, the *anomalous* Hall conductance contributed by each set of Fermi arcs cancels out each other. We thus cannot identify the topological property by probing the anomalous Hall conductance. Fortunately, under magnetic fields along $y$, the two pairs of Fermi arcs both lead to counterclockwise cyclic motions [see appendix A], which give rise to nonzero 3D quantum Hall conductance [see Fig. 4(b)]. In Fig. 4(b), we also plot the numerically computed Hall conductance with respect to the Fermi energy in amorphous lattices, remarkably illustrating the emergence of a quantized plateau at $\overline{\sigma^{\mathrm{H}}_{xy}} = -2e^2/(2h)$ when $E_{\mathrm{F}} \gtrsim 11$. Such a plateau arises from a transition from $\overline{\sigma^{\mathrm{H}}_{xy}} = 0$ to $\overline{\sigma^{\mathrm{H}}_{xy}} = -2e^2/(2h)$ because the two pairs of Fermi arcs in our model are symmetric due to TRS and thus contribute identical Hall conductance.

# 6 Conclusion

In summary, we have theoretically demonstrated the existence of 3D quantum Hall effect in a slab of topological amorphous metals subjected to magnetic fields. We find that while structural disorder significantly broadens the Landau levels such that two neighboring Landau levels overlap with each other, the bulk states are spatially localized except at critical transition energies so that they do not contribute to the Hall conductance. However, due to the topological property of the Landau levels revealed by the Bott index, the edge states arise at distinct hinges on two opposite surfaces, leading to 3D quantum Hall effect. Our results indicate that Weyl band like topology in amorphous materials can be identified through measuring the 3D quantum Hall effect. In appendix B, we also demonstrate the existence of 3D quantum Hall effects in the amorphous system under a tilted magnetic field. Our results suggest that 3D

quantum Hall effects may broadly arise in amorphous Weyl like materials. In fact, 3D quantum Hall effect has been experimentally observed in crystalline semimetal materials, such as $Cd_3As_2$ [36–39]. We thus expect their amorphous counterparts might also exhibit 3D quantum Hall effect.

# Acknowledgements

We thank Y.-B. Yang and B. Wieder for helpful discussions. We also acknowledge the support by Center of High Performance Computing, Tsinghua University.

**Funding information**     The work is supported by the National Natural Science Foundation of China (Grant No. 11974201), Tsinghua University Dushi Program and Innovation Program for Quantum Science and Technology (Grant No. 2021ZD0301604).

# A   Semiclassical explanation of the 3D quantum Hall effect for a time-reversal invariant Hamiltonian on regular lattices

In the main text, we have shown that 3D quantum Hall effect can arise in a time-reversal invariant Hamiltonian $\hat{H}_T$ with double pairs of Weyl points on regular lattices under a magnetic field. In this section, through semiclassical analysis based on the semiclassical equations of motion [40], we will account for why the two pairs of Fermi arcs give rise to nonzero quantum Hall conductance rather than cancel out the contribution from each other as for the *anomalous* Hall conductivity without magnetic fields.

The double pairs of Weyl points and two loops of Fermi arcs on top and bottom surfaces of the Hamiltonian in a regular lattice are schematically shown in Fig. 5(a). For clarity, we project the Fermi arcs on both surfaces onto the same plane. At Weyl points, Fermi arcs on opposite surfaces are connected by chiral Landau levels. The semiclassical equation of motion $\dot{r} = \partial \varepsilon_k / (\hbar \partial k)$ tells us that the real space velocity $\dot{r}$ is perpendicular to the Fermi arc. In our case, $\dot{r}$ points out of the loops formed by Fermi arcs, as indicated in Fig. 5(a). Under a magnetic field along $y$, the electronic wave packets cycle along the loops of Fermi arcs counterclockwise in momentum space, also shown in Fig. 5(a), determined by the equation $\hbar \dot{k} = -e \dot{r} \times B$. Specifically, as illustrated in Fig. 5(b), for an electron starting from the Weyl point $A_{1m}$ on the top surface in momentum space, it will move along the Fermi arc to another Weyl point $B_{1m}$ and then tunnel to $B'_{1m}$ on the bottom surface. Later, the electron travels along the Fermi arc on the bottom surface to $A'_{1m}$ and tunnel back to $A_{1m}$, completing a cyclic motion in momentum space. Correspondingly, in real space, the electron moves from $A_{1r}$ to $B_{1r}$ and then tunnel to $B'_{1r}$ on the bottom surface, as shown in Fig. 5(c). After traveling to $A'_{1r}$ and tunneling back to $A_{1r}$, a cyclic motion in real space is completed. The cyclic behavior of electrons on the other pair of Fermi arcs [between $A_{2m}$ ($A'_{2m}$) and $B_{2m}$ ($B'_{2m}$)] and corresponding real space motions can be extracted in a similar manner. Viewed from the positive direction of the $y$ axis, in both momentum and real space, the cyclic directions of electrons dominated by two loops of Fermi arcs are both counterclockwise. In fact, the same cycling direction for different pairs of Fermi arcs is ensured by the time-reversal symmetry. At the edge, the cyclic motions cannot be finished, giving rise to edge states, as shown in Fig. 5(d). The edge states contributed by the left pair of Fermi arcs are indicated by grey color and those contributed by the right pair are colored green. Under the setup of Hall conductance measurement in Fig. 2(b) in the main text, the two pairs of edge states contribute quantum Hall conductance with the same sign and nonzero total quantum Hall conductance is measurable.

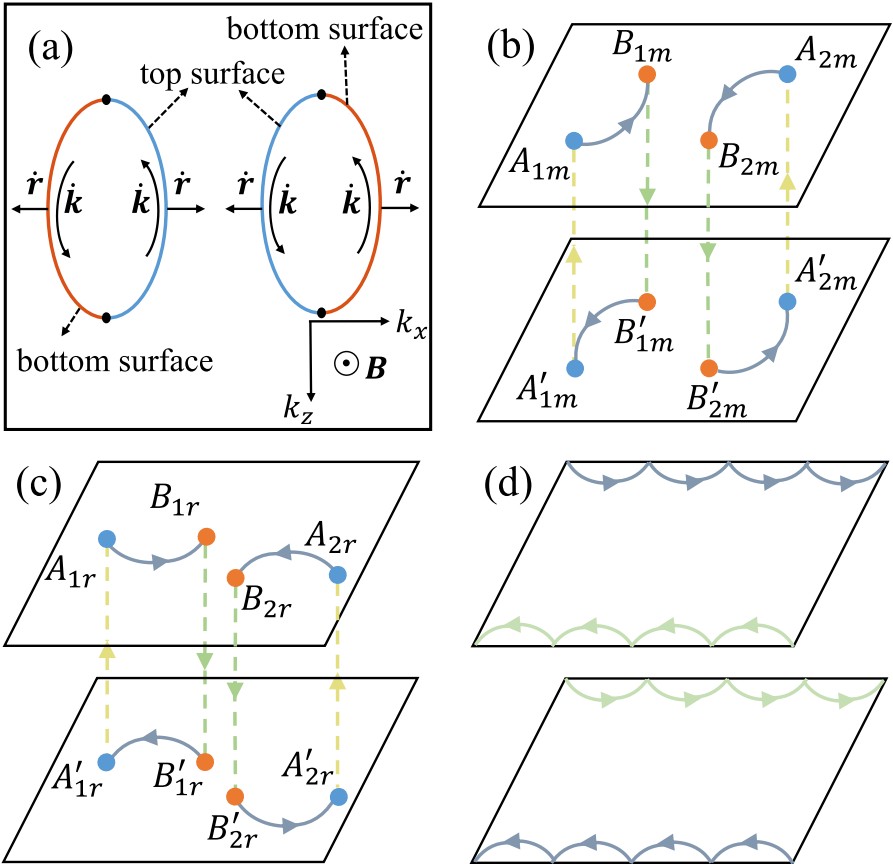

Figure 5: (a) Schematic of Weyl points and Fermi arcs on top and bottom surfaces projected onto a single plane for the Hamiltonian $\hat{H}_T$ on a regular lattice in the main text. The velocities of the center of mass of an electronic wave packet $\dot{k}$ and $\dot{r}$ in momentum and real space, respectively, under a magnetic field in $y$ direction are indicated in the figure. (b) Schematic of cyclic motions of electrons in momentum space where Weyl points are denoted by $A_{im}$ ($A'_{im}$) and $B_{im}$ ($B'_{im}$) with $i = 1, 2$. The chiral Landau levels tunneling from top to bottom surface and vise versa are colored green and yellow, respectively. (c) Schematic of cyclic motions of electrons in real space. (d) Schematic of edge states contributed by the left pair (colored grey) and right pair (colored green) of Fermi arcs.

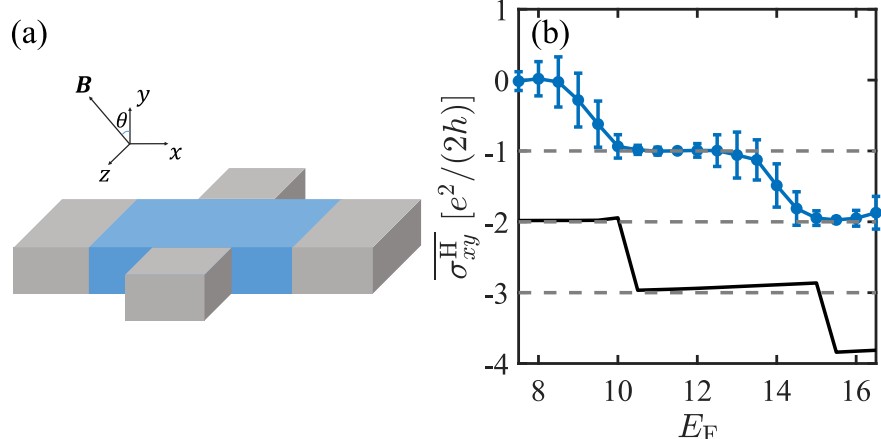

Figure 6: (a) Schematic of the sample under a tilted magnetic field $\boldsymbol{B}$. (b) Configuration averaged Hall conductance $\overline{\sigma_{xy}^{\mathrm{H}}}$ with respect to the Fermi energy $E_{\mathrm{F}}$ for amorphous (the blue line) and regular (the black line) lattices. The sample size is $L_x = 200, L_y = 20, L_z = 40$ and the parameters of the Hamiltonian are the same as in the main text.

## B    3D quantum Hall effect on amorphous lattices under a tilted magnetic field

In the main text, we have demonstrated the existence of 3D quantum Hall effect on amorphous lattices under a magnetic field along $y$. In this section, we will show that it also occurs under a tilted magnetic field. Specifically, we study the Hamiltonian $\hat{H}$ in the main text under a magnetic field $\boldsymbol{B} = (B_x, B_y, B_z)$ lying in the $(y, z)$ plane with a tilting angle $\theta$ from the $y$ axis, i.e. $B_x = 0$ and $B_z = B_y \tan \theta$, as shown in Fig. 6(a). Here, we take $B_y = \pi/22$ and $\theta = 10°$ and the calculated Hall conductance for amorphous (the blue line) and regular (the black line) lattices is displayed in Fig. 6(b). We see that the Hall conductance on amorphous lattices is quantized at $\overline{\sigma_{xy}^{\mathrm{H}}} = -e^2/(2h)$ for $10 \lesssim E_{\mathrm{F}} \lesssim 13.5$ and at $\overline{\sigma_{xy}^{\mathrm{H}}} = -2e^2/(2h)$ for $15 \lesssim E_{\mathrm{F}} \lesssim 16$, showing the existence of 3D quantum Hall effect on amorphous lattices. The Hall conductance plateau at a certain value of $\overline{\sigma_{xy}^{\mathrm{H}}}$ moves to higher Fermi energy $E_{\mathrm{F}}$ compared to the Hall conductance on a regular lattice, attributed to the shifting of Landau levels to the higher energy as illustrated in the main text. We observe that although the component of the magnetic field perpendicular to the top and bottom surface, namely $B_y$ is the same as in the main text, the Hall conductance is different at the same Fermi energy $E_{\mathrm{F}}$ [compared with the blue line in Fig. 2(a) in the main text], implying the angle dependence of 3D quantum Hall effect on amorphous lattices, similar to the regular case [40]. Note that the Hall conductance is not well quantized at $\overline{\sigma_{xy}^{\mathrm{H}}} = -2e^2/(2h)$ for $10.5 \lesssim E_{\mathrm{F}} \lesssim 15$ and at $\overline{\sigma_{xy}^{\mathrm{H}}} = -4e^2/(2h)$ for $15.5 \lesssim E_{\mathrm{F}} \lesssim 16.5$ on the regular lattice due to finite-size effect.

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
