# Peer review of "Three-Dimensional Quantum Hall Effect in Topological Amorphous Metals"

_SciPost Physics_

## Round 2 · Referee Report · Anonymous (Referee 1) · 2023-12-15

Strengths

1 - Novel findings, with an approach motivated by experimental relevance

2 - Multiple methods applied to verify and explain results

3 - Generally clear writing and presentation

Weaknesses

1- The cases with and without time-reversal symmetry could be compared and contrasted more; results are often obtained only for one or the other with no explanation for the choice.

2 - Potentially relevant aspects regarding parameters, in particular the magnitude of the magnetic field, left vague or unexplored.

Report

The main finding of the work appears to be that the presence of magnetic fields can allow for a detection of Weyl bands in amorphous materials even when time-reversal invariance (in the absence of a magnetic field) causes the conductance from those bands to ordinarily cancel out. With recent experiments finding evidence for a 3D quantum Hall effects in crystalline system, and the challenges with measuring amorphous anomalous Hall conductance mentioned in the manuscript's introduction, the findings are quite relevant for current research in the field. The authors find that topology (as calculated by Bott index) and observables (Hall conductivity) match well, and explain the well-defined quantization by considering the localization length of bulk wavefunctions.

Two Hamiltonians are considered, first without and later with time-reversal symmetry. The latter seems potentially more significant as a finding, as the case with broken TRS, although without a magnetic field, was already considered in Ref. [54].

However, many of the results, e.g. the localization lengths and scaling, as well as the tilted magnetic field in the appendix, are only obtained for the TR-breaking case. Even if I do find it plausible the results would not be too dissimilar, an explicit discussion of this is absolutely warranted.

As for the TR-breaking case, given that [54] found a quantized Hall conductance with no external magnetic field, the claim that parameters are chosen "with no loss of generality" is not obvious to me. Intuitively, while the Hamiltonian is different, could not a large enough TR-breaking term even here yield the observed results regardless of any external magnetic field? An elaboration on the effects of the magnitude of B on the observed results would do much to clarify this. This would also add scientific novelty to this part (in taking more of distance from [54]).

There are also some other details that could, in my opinion, improve the manuscript:

  • It is specified that the sample used be thin along y. Whether this was a choice made for computational expedience, or something that is significant to the results, could be mentioned.

  • The LDOS figures show high values in certain locations in the bulk of the system. While this would be expected for individual realizations, it is not immediately obvious why this feature is seen in a sample-averaged quantity, where fluctuations due to random sites should be equally distributed over the bulk. Is it a finite-size effect due to the location of the leads? This could also be mentioned in the text.

  • Given the current status of experimental research into the 3D quantum Hall effect, it could be of interest to consider whether there are any known materials that may be used for studying the case presented here, and if so how relevant parameters in those compare to the model Hamiltonians here.

I do hold that there may be results here of sufficient novelty and relevance to the scientific community to warrant publication, especially with regards to the TR-invariant Hamiltonian. However, the comparison between the separate parts of the work and between this and previous works - chiefly [54] - should be elaborated on in the text. In its current format these aspects are left much too vague.

Requested changes

1 - Add a discussion of to what extent results apply to case with / without time-reversal symmetry

2 - Elaborate on the effects of the magnitude of B on the results.

  • validity: high
  • significance: good
  • originality: good
  • clarity: good
  • formatting: perfect
  • grammar: excellent

Author:  Jiong-Hao Wang  on 2025-03-04  [id 5262]

(in reply to Report 1 on 2023-12-15)

Please find the detailed reply in the attachment.

Attachment:

Reply_referee1.pdf

---

## Round 2 · Referee Report · Anonymous (Referee 2) · 2024-1-3

Strengths

  • potentially novel way of detecting topology in amorphous topological metals

  • clearly written manuscript

Weaknesses

  • the more interesting case of topological semimetal phase protected by the time reversal symmetry is less explored

  • it is not clear whether the results are generic to all time-reversal symmetry protected topological phases

Report

The authors show that the amorphous topological metallic systems may exhibit a 3D quantum Hall effect in presence of the magnetic field. Since this phenomenon can be observed experimentally, the work offers a way to detect/confirm topological features in amorphous metallic systems that do not admit momentum-space topological invariants. This is particularly relevant for Weyl semimetals protected by the time-reversal symmetry for which most of the real-space approaches to calculating the topological invariant fail, with the exception of the spectral localizer (see Schulz-Baldes & Stoiber EPL 136 27001 (2021) and J. Math. Phys. 64, 081901 (2023); Cerjan & Loring PRB 106 064109 (2022); Dixon et al. PRL 131 213801 (2023); Franca & Grushin arXiv: 2306.17117).

I find the manuscript clearly written with interesting results. Provided the authors answer my comments, I would be happy to recommend this work for publication in Scipost Physics Core.

Requested changes

  1. I do not agree with the authors that ARPES cannot be used to probe amorphous systems as it was in fact used in their Ref. 70 to demonstrate experimentally the existence of a topological phase in amorphous Bi2Se3. This is because in amorphous systems, we can still have well defined plane waves of momentum k describing the outgoing electron. The ARPES measures the overlap of these plane waves with the eigenstates of the Hamiltonian, implying that sharp spectral features in ARPES of amorphous systems indicate the presence of states.

  2. In the last line of page 3, the authors set a value for parameter $A$ that I could not find previously defined in the text. Since their Ref. 8 uses a parameter A in the Hamiltonian, and they use $\gamma$ I wonder shouldn't $A$ be replaced with $\gamma$? This should also explain why they never set the value of parameter $\gamma$ in the manuscript.

  3. In Figs. 2(c) and (d), it is very unusual to see that the bulk states remain so visible even after averaging over 100 configurations. Could the authors explain this?

  4. Concerning section 5 that focuses on the time-reversal symmetry protected Weyl semimetals, it is not clear to me whether the results the authors obtain are specific to this model or are generic to this class of systems. It would significantly add to the value of the manuscript if the authors could provide a discussion on this.

  5. In Fig. 6(b), we see that the Hall conductances for crystalline and amorphous systems have a very similar dependence on $E_F$, in comparison with the time-reversal broken case. What would be the reason for this? In addition, I find that having a calculation similar to Fig. 3(a) would enrich the manuscript.

  6. Could the authors calculate the Bott index for Hamiltonian Eq. (6) in presence of magnetic field? It would be a great to show the bulk-boundary correspondence works for this case.

  7. Finally, I wonder whether about the interplay between disorder strength and the magnetic field. In the presence of disorder, does the nonzero Hall conductance appear for any magnetic field strength?

  • validity: high
  • significance: high
  • originality: high
  • clarity: good
  • formatting: good
  • grammar: good

Author:  Jiong-Hao Wang  on 2025-03-04  [id 5261]

(in reply to Report 2 on 2024-01-03)

Please find the detailed reply in the attachment.

Attachment:

Reply_referee2.pdf

---

## Editorial Decision

resubmitted